



# Representing Model Uncertainty for Global Atmospheric CO$_2$ Flux Inversions Using ECMWF-IFS-46R1

Joe McNorton[1], Nicolas Bousserez[1], Anna Agustí-Panareda[1], Gianpaolo Balsamo[1], Margarita Choulga[1], Andrew Dawson[1], Richard Engelen[1], Zak Kipling[1] and Simon Lang[1]

[1]European Centre for Medium-Range Weather Forecasts, Reading, RG2 9AX, UK

*Correspondence to*: Joe R. McNorton (joe.mcnorton@ecmwf.int)

Atmospheric flux inversions use observations of atmospheric CO$_2$ to provide anthropogenic and biogenic CO$_2$ flux estimates at a range of spatiotemporal scales. Inversions require prior flux, forward model and observation errors to estimate posterior fluxes and uncertainties. We use a numerical weather prediction model to diagnose the global forward model error associated

with uncertainties in the initial meteorological state, physical parameterisations and in-model biogenic response to meteorological uncertainty. We then compare the error with the atmospheric response to uncertainty in the prior anthropogenic emissions. Although transport errors are variable, average total column CO$_2$ (XCO$_2$) transport errors over anthropogenic emission hotspots (0.1-0.8 ppm) are comparable to, and often exceed prior monthly anthropogenic flux uncertainties projected onto the same space (0.1-1.4 ppm). Average near-surface transport error at 3 sites (Paris, Caltech and Tsukuba) range from

1.7-7.2 ppm. The global average XCO$_2$ transport error standard deviation plateaus at ~0.1 ppm after 2-3 days, after which atmospheric mixing significantly dampens the concentration gradients. Error correlations are found to be highly flow-dependent, with XCO$_2$ spatiotemporal correlation length scales ranging from 0 km to 700 km and 0 to 260 minutes. Globally, the average model error caused by the biogenic response to atmospheric meteorological uncertainties is small (<0.01 ppm); however, this increases over high flux regions and is seasonally dependent (e.g Amazon January/July: 0.24±0.18

ppm/0.13±0.07 ppm). In general, flux hotspots are well correlated with model transport errors. Our model error estimates, combined with the atmospheric response to anthropogenic flux uncertainty, are validated against 3 TCCON XCO$_2$ sites. Results indicate our model and flux uncertainty accounts for 21-65% of the total uncertainty. The remaining uncertainty originates from additional sources, such as observation, numerical and representation errors, and structural errors in the biogenic model. An underrepresentation of transport and flux uncertainties could also contribute to the remaining uncertainty.

Our quantification of CO$_2$ transport error can be used to help derive accurate posterior fluxes and error reductions in future inversion systems. The model uncertainty diagnosed here can be used in varying degrees of complexity and with different modelling techniques by the inversion community.

## 1 Introduction

Since 1750 global atmospheric $CO_2$ concentrations have increased from 277 ppm (Joos and Spahni, 2008), to 2019 values of 410 ppm (Dlugokencky and Tans, 2019). The initial growth in $CO_2$ was primarily caused by land-use change and then subsequently more by fossil fuel sources. The budget contribution from anthropogenic sources along with existing ocean and biogenic fluxes is difficult to disentangle, both at short (days) and long (decades) timescales. For example, Le Quéré et al. (2018) found a 2008-2017 budget imbalance of 0.5 GtC yr$^{-1}$ caused by uncertainties in the fossil fuel emissions, land-use

change, and land/ocean sink.

Atmospheric inversions are often used to estimate both biogenic and anthropogenic $CO_2$ fluxes at a range of spatial and temporal scales (e.g. Gurney *et al*., 2002; Peylin *et al*., 2013; Lauvaux *et al*., 2016). These inversions typical follow a Bayesian framework whereby prior information is used in an atmospheric transport model, those fluxes and uncertainties are then updated based on comparisons with atmospheric observations. Inversion intercomparison studies show that whilst model

agreement is improving, large differences remain between different inversion systems (Peylin *et al.*, 2013; Le Quéré *et al*., 2018; Gaubert *et al.*, 2019). These are caused by a combination of differences in the prior information, transport model and observation networks used to constrain the fluxes.

Bayesian $CO_2$ inversions require a combined knowledge of the prior uncertainty, model transport uncertainty, measurement error and representation error to provide an accurate estimation of fluxes (e.g., Engelen et al., 2002). Neglecting these

components of uncertainty imposes a hard constraint on the inversion resulting in unreasonable solutions.

Prior fluxes are typically derived from bottom-up process models and observations. The uncertainty can, in part, be estimated by sampling the prior inventory probability distribution function (PDF), perturbing the meteorological data used to force the process models, using ancillary information on uncertainty estimates (e.g. national energy statistics) or a combination of these. Spatial and temporal prior flux error correlation structures can also be considered (e.g. Wu *et al*., 2013). The prior uncertainty

is often only applied to the biogenic fluxes, with an assumed perfect knowledge of the anthropogenic flux, although joint inversions of both biogenic and anthropogenic fluxes require consideration of uncertainties from both.

The observation uncertainty is independent, relatively small and well-known for *in-situ* observations and the application of this uncertainty to an inverse system is straightforward. For satellite observations, spatially coherent biases might influence uncertainties (Basu *et al.*, 2018).

The representation error consists of two components. Firstly, the internal model component, which relates to the model inversion resolution being lower than that of the forward model (see Engelen *et al.*, 2002 for more details). Secondly, the error that arises from spatiotemporal differences between model and observations, for example a point measurement compared to a



model grid box average. This error is expected to reduce as both forward and inverse model resolution increases, and to an extent can be quantified using multi-resolution models (see Agustí-Panareda *et al.*, 2019 for more details).

Here, we investigate the forward transport error and the associated biogenic feedback in an Earth System Model (ESM) context. Model transport error is usually larger than the observation error (Stephens *et al.*, 2007; Law *et al.*, 2008) and often consists of simplified assumptions. Depending on the configuration of the forward model, errors can occur from uncertainty in the initial meteorological conditions, the analysis fields used or in the advection schemes and physical parameterisation of the model.

Uncertainties in the physical parameterisation of land surface and planetary boundary layer schemes can cause errors in the mixed layer (ML) depth, which can lead to errors in the vertical mixing of $CO_2$ (Sarrat *et al.*, 2007; Díaz-Isaac *et al.*, 2018). For $CO_2$, the biogenic flux exchange at the surface correlates with changes in the ML depth making the issue more complex (Denning *et al.*, 1995). When performing inversions using surface observations, an accurate representation and consideration of any uncertainties in vertical mixing is especially important to avoid biases in estimated fluxes (Yi *et al.*, 2004; Denning *et al.*, 2008; Ahmadov *et al.*, 2009). For aircraft and column observations the errors in the vertical mixing may become less important, for example, Verma *et al.* (2017) found inverse flux estimates from aircraft profiles are not sensitive to errors in the ML depth. Similarly, satellite-based inversions, which retrieve total column $CO_2$ ($XCO_2$), are expected to be less sensitive to vertical mixing errors. However, the issue of sensitivity becomes more complex in this case because the $XCO_2$ signal is smaller than the ML signal (Basu *et al.*, 2018). In addition to vertical mixing, advection errors associated with horizontal wind can
result in errors up to 6 ppm (Lin and Gerbig 2005).

CO$_2$ inversions are performed using either an online model, with a full physics scheme used to compute the meteorology, or offline, using analysis transport fields. Online inversions are computationally expensive, require access to a numerical weather prediction (NWP) system and, without the benefit of analysed transport fields, are limited by the accuracy of the physical forecast model. There is the added logistical challenge of reconciling the relatively short NWP assimilation window length
(hours to days) with the typically longer CO$_2$ window length (weeks to years). Typically, online systems have a higher temporal frequency than offline systems, which are limited by the output frequency of the archived analysis fields used. Vertical transport and other sub-grid scale processes, which are missing from the analysis, are computed by offline systems using schemes that are likely to be inconsistent with the original analysis, resulting in further errors (Engelen *et al.*, 2002). Within an online ESM context, biogenic fluxes and surface parameter estimation can be integrated within the inversion system at a
high temporal resolution. The advantages of an online inversion system for the attribution of model uncertainty are investigated here.

Ensembles of transport models are often used to quantify transport uncertainty (e.g. Gurney *et al.*, 2002; Baker *et al.*, 2006; Peylin *et al.*, 2013; Basu *et al.*, 2018). Whilst this represents the variability between models, systematic errors inherent within



those models remain unaccounted for. For example, several models within an ensemble may use the same planetary boundary
layer scheme, resulting in an unrealistic assumption of transport uncertainty. Ensembles using multiple schemes or resolutions
may yield different inverse results (Gaubert *et al.*, 2019), but this does not necessarily mean they provide an accurate
representation of transport uncertainty. Alternatively, multi-physics ensembles with perturbed parametrisations provide a
representation of transport uncertainty caused by parametric uncertainty during the simulation period (Kretschmer *et al.*, 2012;
Lauvaux and Davis, 2014; Díaz-Isaac *et al.*, 2019). The stochastistic representation of model uncertainty required for reliable
ensemble forecasts has been thoroughly researched within the NWP community (e.g. Leutbecher *et al.*, 2017). The ensemble
approach may also consist of models which use forcing data taken from the same analysis product, leading to an underestimate
in the uncertainty associated with the initial conditions and meteorological fields. A representation of uncertainties in initial
meteorological conditions, boundary conditions (for regional models), forcing data and model physics is required to accurately
evaluate transport uncertainty. Numerical uncertainty in models including errors relating to interpolation, diffusion and
advection also contribute to transport uncertainty, although these are not investigated in this study. A complementary approach
to quantify transport uncertainty is to perform direct comparisons with modelled and observed meteorological variables, as
described by Lin and Gerbig (2005).

Here we use an NWP ensemble forecast system, initialised from an ensemble data assimilation (EDA) system, to investigate
transport model uncertainty relating to both the uncertainty in the initial meteorological conditions and in the model physics.
Furthermore, we explore the spatiotemporal variability and flow-dependent error covariances. We perform preliminary
investigations into the biogenic fluxes associated with the meteorological uncertainty, resulting in a more complete model
uncertainty. The biogenic feedbacks here do not account for parametrisation and mapping uncertainties. Finally, we investigate
the signal-to-noise ratio for a prospective $CO_2$ flux inversion system by comparing model uncertainties to the atmospheric
response to anthropogenic emission uncertainties. The combined $XCO_2$ error from model uncertainty and anthropogenic flux
uncertainty is validated against Total Carbon Observing Network (TCCON) observations. If the model uncertainty is
comparable to the model-observation error, as given by a control experiment, then it can be reasoned that the estimated model
uncertainty is a relatively accurate estimation of the true model uncertainty. Other errors not accounted for, for example the
representation error, would further increase this error towards the true model uncertainty.

## 2. Model Setup

We have used version 46R1 of the Integrated Forecasting System (IFS), operated and licenced by the European Centre for
Medium-Range Weather Forecasts (ECMWF). A detailed scientific and technical description of the IFS can be found at
https://www.ecmwf.int/en/forecasts/documentation/evolution-ifs/cycles/summary-cycle-46r1 (last access: 22 September
2019). The IFS primary use is in NWP, although extensions exist for atmospheric $CO_2$ modelling. We used the Ensemble
Prediction System (EPS) component of the Integrated Forecasting System (IFS), detailed in Leutbecher and Palmer (2008), to



simulate 3-D atmospheric $CO_2$ concentrations, given a combination of prescribed and modelled surface fluxes. The EPS is configured to represent both the uncertainty in initial meteorological conditions and in model formulation. The uncertainty in initial conditions were inherited from an operational EDA, where input observations were perturbed with stochastic noise based on a given observation error (Isaksen *et al.*, 2010). In addition to this, both the EPS and EDA use a Stochastically Perturbed Parameterisation Tendencies (SPPT) scheme to represent errors caused by uncertainty in physical parameterisations,

including subgrid-scale processes (Buizza *et al.*, 1999; Leutbecher *et al.*, 2017). Different from the operational configuration of the EPS we start the ensemble members directly from the EDA members instead of adding perturbations to the deterministic analysis. Furthermore, we do not apply singular vector perturbations to the initial conditions.

All simulations were performed globally for January and July 2015 with 137 vertical levels and at ~25km horizontal resolution (TCo399). Instantaneous 3-D model $CO_2$ fields and biogenic fluxes calculated online by CTESSEL, the land surface

component of the IFS (Boussetta *et al.*, 2013; Agustí-Panareda *et al.*, 2014; Agustí-Panareda *et al.*, 2016), were output at hourly frequency. The uncertainty in each simulation is represented by the standard error of a 50-member ensemble, the sampling error resulting from the ensemble size is discussed in the following sections. The 3-D $CO_2$ fields for all ensemble members were initialised using the ECMWF operational product, which is provided under the Copernicus Atmosphere Monitoring Service (Agustí-Panareda *et al.*, 2019). Each month-long ensemble member is comprised of 24-hour forecasts

reinitialised from the operational EDA, with the 3-D $CO_2$ field cycled from the last timestep of the previous forecast. As a result, on the first day of the month the ensemble does not include a representation of the initial atmospheric 3-D $CO_2$ uncertainty; however, the error in initial $CO_2$ concentrations for each forecast is established within the ensemble after a few days. To account for this the first 2 days are discarded from all monthly values provided.

Multiple experiments were performed to identify specific contributions to the total ESM uncertainty. Perturbing the initial

conditions, model physics and the meteorologically dependent biogenic flux, provides a representation of model uncertainty, hereafter, this simulation is referred to as FME. Individually, the uncertainties associated with the initial conditions (IME), the model physics (PME) and the biogenic response to uncertainty in meteorological forcing (BME), were investigated by performing ensemble simulations where only the target component was perturbed. It is important to note that the biogenic uncertainty shown here only represents the biogenic feedback to uncertainties in meteorology and not mapping or process

uncertainty inherent within the model. A simulation, where both the initial meteorological conditions and model physics were perturbed (TME), represents the transport model uncertainty by using offline biogenic emissions from a control experiment. Hereafter, transport model uncertainty is defined as the uncertainty associated with the initial conditions and model physics during the integration, which is typically simplified in inverse modelling studies and model uncertainty includes uncertainty in biogenic fluxes associated with meteorological uncertainty. The biogenic response to errors in the forcing is estimated using

the member specific biogenic fluxes from TME as offline fluxes in BME.



Offline biogenic emissions were broadly consistent with online biogenic emissions in that they were generated using CTESSEL, the only difference is in the frequency. The online biogenic emissions were applied at model timestep frequency (20 minutes), whereas the offline biogenic emissions were input at 3-hour intervals and interpolated across each timestep. Unless otherwise stated offline biogenic emissions were generated using a control forecast. Offline monthly anthropogenic

emissions were generated using EDGAR v4.3.2 (Janssens-Maenhout *et al.*, 2019), extended to 2015 with monthly scaling factors derived from 2010. These were regridded to the model grid from a native 0.1°x0.1° resolution. Daily mean fire emissions were also regridded from 0.1°x0.1° resolution, taken from GFAS (Kaiser *et al.*, 2012). Monthly mean ocean fluxes were taken from Jena CarboScope v1.6 based on the SOCAT data set of $pCO_2$ observations (Rödenbeck *et al.*, 2013). The uncertainties in fire and ocean fluxes are not considered here.

The forward model component of an ensemble-based $CO_2$ flux inversion provides an estimated PDF of atmospheric $CO_2$ based on a signal (prior emission uncertainty) and noise (model uncertainty). To investigate the signal-to-noise ratio relevant for anthropogenic $CO_2$ inversions additional simulations were performed using estimated anthropogenic emission uncertainties and are described alongside all other experiment configurations in table 1 (EXP, PEM and PEA). These estimates are calculated following IPCC guidelines (IPCC, 2006) and will be discussed in detail in a follow-up paper (Choulga *et al.*, in preparation).

Anthropogenic emissions were grouped into six sectors, large powerplants, the remaining energy sector, manufacturing, transport, settlements and other. National uncertainties for annual and monthly emissions are strongly sector and country dependent, ranging from annual transport uncertainties of ~4% for numerous developed nations to monthly other sector uncertainties of ~330% for The Democratic Republic of the Congo. Aviation emission were used as 3-D profiles but remained unperturbed in these simulations.

The uncertainties used here are thought to be relatively modest considering the timescales being investigated. Data availability of several aspects of anthropogenic uncertainties currently limit our ability to diagnose a reasonable atmospheric $XCO_2$ response signal at short timescales. For example, daily uncertainties, which would be required for high temporal frequency flux inversions, are expected to be considerably larger than monthly uncertainties. This would provide, in principle a larger signal. Additionally, a lack of prior information prevented the consideration of uncertainty correlations in prior fluxes. Finally,

the diurnal variability in emissions, which is likely to influence the modelled atmospheric response to anthropogenic emissions, is not considered. The missing information in prior uncertainties of anthropogenic fluxes leads to an underestimation of the flux signal, and as a result the signal-to-noise ratio.

| Name | Initial Conditions | Physics | Biogenic Emissions | Anthropogenic Emissions | Error Information |
| --- | --- | --- | --- | --- | --- |
| IME | EDA | SPPT off | Offline | Fixed | Initial meteorological |





| PME | Control | SPPT on | Offline | Fixed | Model physics |
| TME | EDA | SPPT on | Offline | Fixed | Transport |
| BME | Control | SPPT off | Offline-FME | Fixed | Biogenic feedback |
| FME | EDA | SPPT on | Online | Fixed | Model (noise) |
| PEA | Control | SPPT off | Online | Perturbed Annual Error | Anthropogenic emission (signal) |
| PEM | Control | SPPT off | Online | Perturbed Monthly Error | Anthropogenic emission (signal) |
| EXP | EDA | SPPT on | Online | Perturbed Monthly Error | Full PDF (signal and noise) |

**Table 1. Configuration of model experiments used for attribution of model uncertainty and the signal-to-noise ratio for atmospheric CO₂ inversions. The control denotes the control member of the EDA.**

**3. Observations**

We used atmospheric $XCO_2$ measurements from the Total Carbon Column Observing Network (TCCON) (Wunch *et al*., 2011) to evaluate the combined forward model error and the atmospheric response to anthropogenic flux uncertainties. Assuming the 50-member ensemble accurately represents the atmospheric $CO_2$ PDF accounting for all uncertainties, the standard error in EXP should be comparable to the model-observation error. However, the total error is expected to under-represent the model-

observation error because some uncertainties were either missing or underestimated by the ensemble. For example, the representation error is not present in our ensemble and the prior anthropogenic flux uncertainty is based on monthly estimates and not weekly or daily values.

Here, we focus on model uncertainty relative to prior anthropogenic flux uncertainty. Therefore, 3 TCCON sites with nearby anthropogenic sources and with available data for 2015 were selected for evaluation, Paris (Té *et al*., 2017), Caltech (Wennberg

*et al*., 2017) located near Los Angeles and Tsukuba (Morino *et al*., 2017) near Tokyo. Sounding-specific TCCON averaging kernels were applied to interpolated model output for direct model-observation comparisons.



## 4. Results

### 4.1 TCCON site specific error representation

All results shown are taken from the January 2015 simulations; results from the July simulations, although discussed here, are
shown in the supplementary material. The relative contribution to total $XCO_2$ variability from the uncertainties in initial
meteorological conditions, model physics and biogenic feedback, as well as the atmospheric response to prior anthropogenic
uncertainty is shown to be location and time dependent (Figure 1 and S1; for illustration purposes only the first 5 days are
shown). After 2-3 days the total error stabilizes, caused by the impact of atmospheric diffusion (Figure 2 and S2). All monthly
averages are calculated after an initial 2-day spin-up.

Over Paris the initial meteorology (IME) and model physics (PME) errors are the largest individual components of the total
$XCO_2$ variability for January, with averages of 0.12±0.07 ppm and 0.09±0.05 ppm respectively. The combined average
transport error (TME) is 0.15±0.08 ppm, with a maximum of 0.61 ppm. The biogenic feedback (BME) errors are small
(0.01±<0.01 ppm). The average atmospheric $XCO_2$ variation associated with annual anthropogenic flux uncertainties (PEA)
is relatively small (0.05±0.04 ppm std dev); however, using monthly uncertainties (PEM) this variability increases slightly
(0.06±0.05 ppm std dev).

Average initial meteorological error (0.16±0.10 ppm) and model physics error (0.19±0.15 ppm) also dominate the total error
over Tsukuba in January, with a combined average transport error of 0.24±0.16 ppm (maximum 1.01 ppm). The biogenic
feedback errors are again smaller in comparison (<0.01±<0.00ppm). Monthly and annual anthropogenic emission uncertainties
consistently produce smaller errors than the total transport error, 0.09±0.09 and 0.03±0.02, respectively.

Over Caltech the January average variability in the atmospheric response to annual anthropogenic emission uncertainties
(0.13±0.13ppm) is lower than that from the initial meteorological error (0.41±0.41 ppm), the model physics error (0.29±0.27
ppm) and the combined transport error (0.50±0.45 ppm - maximum value of 2.55 ppm). Conversely, the monthly anthropogenic
uncertainties produce the largest average error in atmospheric $XCO_2$ (0.61±0.47 ppm). The average biogenic feedback error is
small (0.01±0.01 ppm). Variability between the three sites is a result of multiple factors including nearby fluxes, regional
atmospheric transport and orography. The minor impact of the biogenic feedback, caused by meteorological uncertainty, results
in the FME and TME errors being almost identical at all 3 sites

July simulations show comparable model transport errors to January over Paris (0.15±0.06), decreases over Caltech (0.23±0.10
ppm) and increases over Tsukuba (0.38±0.23 ppm); showing site specific seasonal variability (S1 and S2). The biogenic
feedback error increased over all three sites in July (Paris: 0.02±0.01 ppm, Caltech: 0.02±001 ppm and Tsukuba: 0.04±0.03
ppm) due to northern hemisphere summer. This remains smaller than the transport and anthropogenic uncertainty response but
is no longer negligible. The July spread in the atmospheric response to monthly anthropogenic flux uncertainties is increased



over Paris (0.08±0.05 ppm) and Tsukuba (0.38±0.2 ppm), when compared to January. Over Caltech (0.47±0.19 ppm) the same error is reduced for July.

There is no clear diurnal cycle in the column transport error at any of the 3 stations, January midnight averages at Paris (0.15±0.08 ppm), Caltech (0.48±0.47 ppm) and Tsukuba (0.29±0.18 ppm) are all comparable to midday averages (0.15±0.07 ppm, 0.46±0.32 ppm and 0.25±0.18 ppm, respectively). For July, only Caltech exhibits a slight diurnal cycle, with midday averages of 0.29±0.13 ppm and midnight averages of 0.18±0.05 ppm. Over Caltech in July, a diurnal cycle is found in the atmospheric $XCO_2$ error as a response to both the biogenic feedback uncertainty and anthropogenic flux uncertainty, with midday averages of 0.02±0.01 ppm and 0.73±0.30 ppm, respectively, and midnight averages of 0.01±0.01 ppm and 0.43±0.18 ppm. Without a diurnal cycle in anthropogenic fluxes, this would suggest the diurnal meteorological variability causes the observed difference in model error as the magnitude in prior flux and error remains the same for both night and day. Summertime diurnal variability over Caltech has previously been attributed to the sea-mountain breeze, where $CO_2$ enhanced air masses peak in the afternoon before reducing again in the evening (Agustí-Panareda *et al.*, 2019). These enhancements cause an increase in atmospheric $CO_2$ gradients, resulting in an increased transport error. Diurnal variability in emissions is expected to increase the diurnal signal in the atmospheric transport error, with typically lower night-time emissions resulting in lower transport model errors; however, we have not tested this hypothesis here.

Flux inversions, more specifically posterior error reduction, depend on the signal-to-noise ratio, where the atmospheric response to prior flux uncertainty is the signal and the remaining errors represent the noise. As previously mentioned, we underestimate the noise here by only accounting for some model uncertainty. Using annual anthropogenic uncertainties to perturb January fluxes generates an average signal-to-noise ratio, after a 2-day spin-up, of 0.38±0.37, 0.27±0.16 and 0.20±0.17 at Paris, Caltech and Tsukuba, respectively (Figure 2). Over Caltech and Tsukuba, the ratio does not exceed 1 for the whole of January, and only exceeds 1 for 8% of the month over Paris. Using monthly anthropogenic uncertainties, the signal-to-noise ratio over Paris and Tsukuba after a 2-day spin-up increases to an average of 0.54±0.37 and 0.36±0.21, exceeding 1 for 9% and 1% of the month, respectively. Over Caltech this increases to a ratio of 1.02±0.68, exceeding 1 for 44% of the month. The average signal-to-noise ratio increases at all 3 sites in July to 0.61±0.42 ppm over Paris, 2.48±0.93 ppm over Caltech and 0.94±0.48 ppm over Tsukuba (S2). The ratio exceeds 1 for 11% of the month over Paris, >99% over Caltech and 38% over Tsukuba. It is reasonable to assume that the uncertainties, and therefore the signal-to-noise ratio, will increase by a similar order of magnitude from monthly to daily uncertainties as the increase seen here from annual to monthly uncertainties; however, no data are currently available for daily anthropogenic flux uncertainties.

To evaluate the accuracy of the total error in $XCO_2$ (model uncertainty and atmospheric response to anthropogenic flux uncertainty) the standard error across ensemble members is compared to the control model-observation error from TCCON (Figure 1 and S1). For January, the mean centred model-observation errors are found to be 1.41 ppm at Caltech and 0.54 ppm at Tsukuba, compared to EXP total model uncertainties (transport, biogenic feedback and monthly anthropogenic emission





uncertainty) of 0.69±0.52 ppm and 0.27±0.19 ppm, respectively. There are no available TCCON data available over Paris for

January 2015, the EXP uncertainty over Paris is 0.16±0.06 ppm. For July, the model-observation errors are 0.92 ppm, 0.90

ppm and 1.84 ppm for Paris, Caltech and Tsukuba, respectively, compared to EXP uncertainties of 0.19±0.07 ppm, 0.60±0.23

ppm and 0.56±0.31 ppm. This would suggest that depending on the time and location, the uncertainties explored here account

for 21-65% of the total model uncertainty. As previously mentioned, the monthly uncertainty estimates used here are an

underestimation of the uncertainties at the short timescales being investigated here (hourly or daily). It should also be noted

that additional sources of model-observation variability, such as observation errors, the representation error, numerical errors

and biogenic flux errors relating to both processes and mapping are not considered in these values. Our results show these

additional uncertainties are not negligible and need to be accounted for in addition to the uncertainties derived here.

The vertical error structure for each ensemble configuration at the 3 TCCON sites over a 24-hour period shows column

variability (Figure 3). For all 3 sites individual errors are typically largest near the surface, where the $CO_2$ gradients are the

largest. Over Paris both components of the transport error are noticeable in the mid-troposphere with some model levels

exceeding 1 ppm errors for both initial meteorological and model physics errors individually. On average the near-surface

(~100m) transport error over Paris is 1.7±2.7 ppm, with a maximum of 17.6 ppm. Over Caltech noticeable transport errors are

typically found in the lower troposphere. The average near-surface error is 7.2±6.2 ppm, with a maximum of 21.8 ppm. Over

Tsukuba the initial meteorological condition error is detectable not only near the surface but also in the mid-to-upper

troposphere (~300hPa), with averages of 0.41±0.21 ppm. Near-surface average transport errors are 2.2±2.8 ppm, with a

maximum of 16.6 ppm.

The near-surface, which is typical used for *in-situ* based inversions, average signal-to-noise ratios for monthly anthropogenic

uncertainties are 1.4±0.5, 0.8±0.7 and 0.4±0.2 over Paris, Caltech and Tsukuba, respectively. The ratio exceeds 1 for 78% of

the time over Paris but less frequently over Caltech (36%) and Tsukuba (0%).

All 3 sites do not exhibit a diurnal cycle in the near-surface transport error. For each site the difference in error between day

and night is less than 10%. This assumes the EDA and SPPT accurately represent transport error by perturbing the boundary

layer physics. These results underestimate the diurnal cycle in the transport error by not accounting for diurnal variability in

emissions.

### 4.2 Global and regional model uncertainty

The global XCO₂ uncertainty resulting from uncertainties in emissions, biogenic feedback and transport, which includes both

initial conditions and physics, is found to be spatially and temporally varying (e.g. January 2015 shown by Figure 4). As

expected, the atmospheric XCO₂ signal from monthly anthropogenic emission uncertainties is largest over emission hotspots

in Eastern China, with smaller signals over North America, Europe and the Middle-East (Table 2 and 3). The global average

error for both January and July 2015 is relatively small, 0.01±0.00 ppm, with maximum local instantaneous XCO₂ errors





reaching 9.2 ppm. The error is expected to increase further with uncertainties applied at the hourly or daily timescale, as these currently unavailable values would be larger than both monthly and annual uncertainties.

The $XCO_2$ biogenic feedback error from atmospheric model uncertainty is largest over regions with a high net ecosystem exchange, e.g. The Amazon (January: $0.16\pm0.08$ ppm and July: $0.06\pm0.06$ ppm) and Southern Africa (January: $0.13\pm0.07$ ppm and July $0.05\pm0.07$ ppm). These are also areas with large atmospheric gradients. The high values in southern hemisphere

summer suggest a seasonal cycle in the biogenic feedback error. Globally the average biogenic feedback error is smaller ($<0.01$ ppm) in January and increases slightly in July ($0.02\pm0.00$ ppm), following the seasonal dependence of biogenic fluxes.

The error in atmospheric $XCO_2$ caused by transport model uncertainties correlates with the error caused by both the anthropogenic uncertainties and biogenic feedback uncertainties, as these are the regions with the largest fluxes and as a result, the largest gradients. The globally averaged $XCO_2$ error resulting from the initial model error, model physics error and

combined transport error is $0.06\pm0.00$ ppm, $0.09\pm0.00$ ppm and $0.10\pm0.01$ ppm respectively. Over regions with a high biogenic flux the average transport error further increases, e.g. The Amazon (January: $0.24\pm0.18$ ppm and July: $0.20\pm0.15$ ppm) and Southern Africa (January: $0.30\pm0.26$ ppm and July: $0.18\pm0.21$ ppm). The transport error in these regions exhibits a similar seasonal cycle to the biogenic feedback error, most likely caused by the increased flux in southern hemisphere summer. The increase in transport error is also evident over regions with a high anthropogenic flux (Table 2 and 3). The average transport

model error over these hotspots is similar in July ($0.32\pm0.17$ ppm) as in January ($0.32\pm0.22$ ppm). Considering most of the sites are in the northern hemisphere this would suggest there is little or no seasonal variability in the average transport error over anthropogenic hotspots, although certain hotspots show some seasonal variability (e.g. Los Angeles). The maximum transport error for all times and locations is 9.2 ppm, although globally for individual grid cells and times the error only exceeds 0.5 ppm for ~1% of the time.

Flux inversions, more specifically posterior error reduction, depend on the signal-to-noise ratio, where the atmospheric response to prior flux uncertainty is the signal and the remaining errors represent the noise

The signal-to-noise ratio using monthly and annual anthropogenic uncertainties is location and time dependent, shown in figure 5 and for various emission hotspots in table 2 and 3. After the initial 2-3 days this ratio is typically below 1 when using prior annual anthropogenic uncertainties, with exceptions over Eastern Asia and the Middle East. For prior monthly uncertainties,

large parts of North America, Europe, Asia and some southern hemisphere hotspots consistently exceed 1. Further work is required to investigate more robust daily, or even hourly, uncertainty estimates for each sector, which is relevant for posterior error reductions at high temporal frequencies. The increased uncertainty in fluxes at higher temporal resolution will result in a more accurate total error, increasing the signal-to-noise ratio, resulting in increased posterior error reductions.



| Location | Transport Error (ppm) | Transport Error (min-max, ppm) | Emission Signal (ppm) | Emission Signal (min-max, ppm) | Signal-to-Noise Ratio |
|---|---|---|---|---|---|
| Johannesburg | 0.24±0.08 | 0.10-0.62 | 0.19±0.07 | 0.10-0.40 | 0.79±0.34 |
| London | 0.12±0.03 | 0.05-0.22 | 0.05±0.02 | 0.02-0.15 | 0.39±0.17 |
| Los Angeles | 0.55±0.43 | 0.06-2.23 | 0.91±0.43 | 0.26-1.97 | 1.66±1.16 |
| Moscow | 0.19±0.11 | 0.05-0.71 | 0.23±0.09 | 0.12-0.65 | 1.23±0.76 |
| New York | 0.15±0.08 | 0.05-0.48 | 0.19±0.09 | 0.06-0.47 | 1.29±0.72 |
| Riyadh | 0.14±0.10 | 0.06-0.81 | 0.28±0.13 | 0.11-0.75 | 2.07±0.77 |
| Seoul | 0.19±0.13 | 0.05-0.86 | 0.21±0.15 | 0.03-0.79 | 1.09±0.49 |
| Shanghai | 0.65±0.57 | 0.15-3.75 | 1.44±0.63 | 0.60-4.29 | 2.20±0.97 |
| Singapore | 0.22±0.07 | 0.12-0.56 | 0.09±0.03 | 0.04-0.18 | 0.39±0.14 |
| Tokyo | 0.79±0.95 | 0.09-5.50 | 0.28±0.27 | 0.04-1.38 | 0.36±0.24 |
| Kendal* (RSA) | 0.33±0.15 | 0.08-0.88 | 0.15±0.05 | 0.07-0.29 | 0.44±0.20 |
| Waigaoqiao* (CHN) | 0.42±0.28 | 0.14-1.27 | 0.74±0.63 | 0.15-2.57 | 1.77±0.81 |
| Neurath* (DEU) | 0.14±0.07 | 0.06-0.59 | 0.06±0.03 | 0.02-0.18 | 0.41±0.22 |

**Table 2. Average, minimum and maximum total column model CO$_2$ error statistics for the transport model error and the atmospheric response to monthly emission uncertainties (signal), and the signal-to-noise ratio for various emission hotspots for January 2015. Results are calculated from the 50-member IFS ensemble. * Denotes large power stations.**

| Location | Transport Error (ppm) | Transport Error (min-max, ppm) | Emission Signal (ppm) | Emission Signal (min-max, ppm) | Signal-to-Noise Ratio |
|---|---|---|---|---|---|





| | | | | | |
|---|---|---|---|---|---|
| Johannesburg | 0.18±0.11 | 0.05-0.69 | 0.26±0.18 | 0.06-0.87 | 1.64±0.91 |
| London | 0.16±0.06 | 0.05-0.36 | 0.05±0.02 | 0.02-0.11 | 0.34±0.17 |
| Los Angeles | 0.18±0.06 | 0.05-0.37 | 0.49±0.29 | 0.11-1.48 | 2.78±1.23 |
| Moscow | 0.25±0.14 | 0.08-0.70 | 0.23±0.12 | 0.10-0.76 | 1.01±0.45 |
| New York | 0.36±0.13 | 0.16-0.78 | 0.38±0.20 | 0.06-1.11 | 1.06±0.43 |
| Riyadh | 0.14±0.10 | 0.04-0.59 | 0.11±0.07 | 0.04-0.40 | 0.87±0.33 |
| Seoul | 0.39±0.17 | 0.14-0.85 | 0.43±0.20 | 0.07-0.85 | 1.16±0.40 |
| Shanghai | 0.67±0.11 | 0.05-3.29 | 1.16±0.18 | 0.06-3.14 | 2.32±0.59 |
| Singapore | 0.24±0.09 | 0.11-0.53 | 0.21±0.06 | 0.09-0.37 | 0.96±0.29 |
| Tokyo | 0.61±0.38 | 0.16-2.60 | 0.48±0.30 | 0.11-1.49 | 0.93±0.58 |
| Kendal* (RSA) | 0.32±0.32 | 0.06-1.72 | 0.16±0.09 | 0.05-0.44 | 0.74±0.44 |
| Waigaoqiao* (CHN) | 0.42±0.33 | 0.09-1.88 | 0.52±0.50 | 0.07-2.40 | 1.19±0.66 |
| Neurath* (DEU) | 0.23±0.15 | 0.06-0.98 | 0.09±0.06 | 0.02-0.29 | 0.39±0.18 |

**Table 3. Average, minimum and maximum total column model $CO_2$ error statistics for the transport model error and the atmospheric response to monthly emission uncertainties (signal), and the signal-to-noise ratio for various emission hotspots for July 2015. Results are calculated from the 50-member IFS ensemble. * Denotes large power stations.**

**4.3 Impact of ensemble size**

After 2-3 days the global average transport model error reaches a steady-state, where the model error growth balances with the atmospheric mixing caused by $CO_2$ gradients (Figure 6). After which, the global transport model error remains approximately 0.1 ppm for all ensemble sizes. Globally, as ensemble size tends toward 50, the error across all ensemble members converges.

Here, as a first guess, we investigated the required ensemble size to adequately represent the prior $XCO_2$ PDF, using multiple

sizes available. The model error is within 5% of the 50-member ensemble error for ensemble sizes over 40, 39 and 43 for Paris, Caltech and Tsukuba, respectively (Figure 6). Ensemble sizes <40 provide model error approximations that may not be suitable



for use in inversions. Computational cost currently limits the use of larger ensemble sizes, and optimum ensemble size investigations indicated the 50-member may provide an adequate sample for meteorological errors (Leutbecher et al., 2017), although $CO_2$ poses more specific challenges and requirements.

To investigate the suitability representing the transport error with a gaussian PDF, ensemble members were binned into 0.05 ppm bins and a non-linear least-square fit was applied to provide an estimated gaussian fit for a PDF with 3 terms; $A_0$, $A_1$ and $A_3$.

$$f(x) = A_0 e^{-\frac{\left(\frac{x-A_1}{A_2}\right)^2}{2}} \qquad (1)$$

Assuming the prior PDF is gaussian, results show that ensemble sizes ≤50 can fail to represent a suitable distribution and
contain spurious noise. Over Paris and Caltech, a gaussian distribution is relatively well captured by a 50-member ensemble; however, for Tsukuba either more ensemble members are required or the PDF is not gaussian.

### 4.4 Error correlation

The noise generated by small ensemble sizes creates spurious spatial and temporal error correlations in the $XCO_2$ transport error (Figure 7). This localisation problem is typically addressed by limiting the distance of correlations considered within the
inversion (e.g. Miyazaki *et al*., 2011) or by applying a decay function (e.g. Chatterjee *et al*., 2012). Here we propose that temporal filtering, as shown by artificially creating a 150-member ensemble using neighbouring times from a 50-member ensemble, could be used to reduce spurious error correlations. This is only applicable with suitably high frequency model data. By filtering a small ensemble (10-member) using time smoothing and finding the best fit to a 50-member ensemble, it is typically found that a 2 hour smoothing is optimum with our model setup ($T_{-1}$,$T_0$,$T_{+1}$). The optimum filter length is however,
location and time dependent.

For a given location we assume surrounding $XCO_2$ error correlation values, which are both part of the spatial extent of the plume and greater than the derived e-folding correlation length scale ($R > e^{-0.5}$), represent a non-spurious correlation. The maximum distance of these correlations from the artificially generated TME 150-member ensemble can range between maximum distances of 30 km to 520 km over Paris (Figure 7). Over Caltech and Tsukuba these range from 0 km to 230 km
and 30 km to 700 km, respectively. The flow-dependency suggests that a predefined distance for the correlation filter might limit the available useful information within the inversion system, even when the filter is spatially varying.

For a given time and location, assuming a gaussian error correlation structure may cause an underestimation or overestimation of the correlation length scale, depending on direction (Figure 8 and S3). For most situations, regardless of location, the shortest





correlation length scale is close to the average correlation length in all directions, however the downwind correlation length
scale is typically around twice as far.

For January, the time and direction averaged error correlation length scale, assuming a gaussian distribution, varies across all
3 sites (Paris 67±24 km, Caltech 17±16 km and Tsukuba 59±26 km). In July, over Paris and Tsukuba, the average correlation
length scale is reduced to 61±22 km and 35±16 km, respectively, whereas there is a slight increase over Caltech to 26±14 km.
The large decrease in correlation length scale detected over Tsukuba in summer may be a result of dominant mesoscale
biogenic fluxes in the region during summer months masking the plume from anthropogenic hotspots. The variability in
average correlation length scale is reduced at all three sites during northern hemisphere summer, which is also likely to be the
result of a more active background biogenic flux limiting the maximum spatial extent of the signal from anthropogenic
hotspots. Seasonal variability in local meteorological systems is also likely to cause observed changes in the correlation length
scales derived. At all 3 locations the average error correlation length scale in all directions varies considerably with time,
suggesting flow-dependent information is required and no single length scale should be used (Figure 8 and S3).

The average error correlation in both time and space simultaneously is also considered, again using a simplistic gaussian
assumption (Figure 8 and S3). This shows the time component of the average error correlation varies with location, with an
average time correlation length scale decreasing with distance.

For January Paris (80 minutes) and Tsukuba (150 minutes) both show a relatively short average time correlation length scale
but a long spatial length scale, whereas the Caltech (260 minutes) has a longer time correlation length scale and shorter spatial
length scale. For July the correlation length scale increases over both Paris (120 minutes) and Tsukuba (170 minutes), with
decreases over Caltech (160 minutes). Differences between locations and seasons are caused by changes in fluxes, meteorology
and orography. For instance, over Caltech, shorter spatial correlations and longer time correlations results from the impact of
the Los Angeles basin, which reinforces air stagnation during winter. This effect is less pronounced during the summer due to
the presence of stronger sea breezes.

## 5. Discussion

We have performed multiple ensemble simulations of $CO_2$ using an online NWP model to quantify sources of atmospheric
model uncertainty. We have individually diagnosed the relative contribution of uncertainties from the initial meteorological
state and model physics to the total transport error. This work can be used to inform future atmospheric flux inversion studies
on the spatiotemporal variability of model transport error, which is typically lacking. By utilising the online capability of the
ESM, we have also diagnosed the biogenic flux feedback error associated with uncertainties in atmospheric meteorology. We
have performed ensemble simulations using perturbed anthropogenic emissions to investigate the signal-to-noise ratio, which
provides a first assessment of the posterior error reductions in an anthropogenic inversion system. Finally, we have diagnosed



error correlations and correlation length scales at selected sites. To evaluate the diagnosed error, the results were validated at
3 TCCON sites. The ensemble derived uncertainties found here will be used to model transport errors in a proposed future
operational global $CO_2$ monitoring system being developed as part of the $CO_2$ Human Emissions project.

The transport error is shown to be spatiotemporally varying and is largest near biogenic and anthropogenic flux hotspots.
Transport errors over anthropogenic flux hotspots are on average 0.1-0.8 ppm and 0.1-0.7 ppm for January and July,
respectively. This transport error is comparable to uncertainties in the prior monthly anthropogenic emissions projected onto
the observation ($XCO_2$) space over the same regions (January: 0.1-1.4 ppm and July: 0.1-1.2 ppm). However, since the
proposed future monitoring system will be based on prior flux uncertainties associated with higher temporal resolutions than
those used here (daily/hourly), a significant increase in the signal-to-noise ratio is expected. The estimation of high-frequency
transport error covariance structures is essential to ensuring reliability of the future inversion system. With potential future
improvements to bottom-up flux estimations the signal-to-noise ratio may further decrease in the future, decreasing the
posterior error reduction values that could be expected from such a system. The spatial and temporal variability of errors and
resulting signal-to-noise ratios are influenced by neighbouring hotspots, local orography and meteorological variability. Our
findings, on a global scale, agree well with the regional study of Chen *et al.* (2019).

Atmospheric $CO_2$ transport error initially grows and then plateaus after 2-3 days, depending on the location. After this time
the error growth from uncertainties in transport balance out with the atmospheric $CO_2$ mixing, resulting in a globally averaged
transport error of ~0.1 ppm.

A noticeable transport error is identified in both the near-surface model levels and in the total column $CO_2$. As a result, it is
likely to impact both satellite and surface based atmospheric inversions. These results highlight the importance of including
detailed transport error within atmospheric $CO_2$ inversions, as most previous studies either ignore or use a simplistic
representation of model transport error, leading to over-confidence in results. The near-surface errors found here at three sites
(1.7-7.2 ppm) are comparable to the 3-4 ppm errors found by Díaz-Isaac *et al.* (2018).

The atmospheric $CO_2$ error caused by the biogenic feedback error as a response to uncertainty in meteorology is found to be
small, however, in regions of high net ecosystem exchange this value increases to an average of 0.16 ppm and requires
consideration for high precision atmospheric inversions in those regions. Both the atmospheric response to prior anthropogenic
emission uncertainties and the biogenic feedback errors are found to be seasonally dependent for some locations caused by
seasonal changes in flux and meteorology. This also results in seasonal variability in the model transport error over regions of
high net ecosystem exchange. The error associated with biogenic fluxes shown here does not account for uncertainties in the
biogenic model or ancillary information (e.g. mapping or plant functional type).

Validation performed with TCCON observations suggests the uncertainty derived in model $XCO_2$ from transport uncertainty, anthropogenic flux uncertainty and biogenic feedback to meteorological uncertainties accounts for 21-65% of the total model

uncertainty, depending on time and location. An underrepresentation of anthropogenic flux uncertainty, by using monthly and not higher temporal resolution uncertainties, and other factors including observation errors, numerical errors, the representation error, missing biogenic processes and biogenic mapping errors make up the remaining model uncertainty. These remaining uncertainties are not negligible, for example a previous study showed over the same Caltech site as used in this study, the model representation error is typically 2 ppm for January (Agustí-Panareda *et al.*, 2019). Future studies should aim to quantify

these additional aspects of model uncertainty.

The 50-member ensemble used here is shown to provide a reasonable estimate of the prior PDF; however, for some regions, ensemble sizes larger than 50 members may be required. The computational cost of sufficiently large ensemble sizes to describe the spatial error structures could potentially be overcome by appropriate filtering techniques of smaller ensemble sizes (e.g. Lauvaux *et al.*, 2019).

Spurious noise is evident in the transport error correlation structure of a 50-member ensemble, to address this issue and prevent further computational costs we apply a simple time filtering to artificially increase the member size to 150 members. Error correlation structures are shown to be strongly flow-dependent. Using a simplified gaussian assumption the average correlation length scale values are found to be between 0 and 700 km in distance and 0 and 260 minutes in time, with a seasonal dependence based on changes in flux and meteorology.

The transport uncertainty diagnosed here highlights the importance of accounting for all sources of model error when performing inversions. Our results are derived using an online NWP system; however, our findings can be used in various levels of complexity to inform future $CO_2$ offline inversions at both the regional and global scale. It should be noted that whilst these uncertainties can be used in an offline system, several additional errors would also need to be considered, including interpolation errors and inconsistencies between transport parameterisations. The model error PDF, although reasonably well

represented by the 50-member ensemble, requires either additional ensemble members or suitable selection techniques (e.g. Díaz-Isaac *et al.*, 2019), which requires further investigation.

*Code availability.*

The IFS source code is available subject to a licence agreement with the ECMWF; see also Leutbecher et al. (2017) for details on the ensemble model description and specific details of the code relevant to this study, including use of the EDA and SPPT.

ECMWF member-state weather services and their approved partners will be granted access. Components of the IFS code relevant to this study (e.g. SPPT), without modules for data assimilation, are also available for educational and academic purposes as part of the OpenIFS project (https://software.ecmwf.int/wiki/display/OIFS/OpenIFS+Home, last access: 09

December 2019). Technical developments specifically related to work detailed here are available upon request, please contact joe.mcnorton@ecmwf.int. The specific code relevant to this study for emissions perturbations based on given log-normal
uncertainties is available at https://bitbucket.org/joemcnorton/mcnorton_gmd/ (last access: 09 December 2019).

*Data availability.* Model data are available online through the ECMWF Meteorological Archive and Retrieval System (MARS) catalogue, but access may be limited. Model output data are available upon request to joe.mcnorton@ecmwf.int

*Author contributions.* The simulations were performed by JM with coding developments from AAP, AD, ZK and SL. The experimental design was devised by JM, NB, AAP, GP, RE and SL. Emission inventories were compiled by JM and AAP,
with uncertainties and budgets calculated by JM and MC. The manuscript was prepared by JM with analysis interpretation from NB and, input and feedback from AAP, GP, NB, RE and SL.

*Competing interests.* The authors declare that they have no conflict of interest.

*Financial support.* This research has been supported by The CHE project. The CHE project has received funding from the European Union's Horizon 2020 research and innovation programme under grant agreement No 776186.

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





**Figure 1. IFS model XCO₂ (ppm) variability over three TCCON sites for 50-member ensemble for 1-5ᵗʰ January 2015 from uncertainties in model transport (first row), biogenic feedback from meteorological uncertainty (second row), monthly uncertainties in anthropogenic emissions (third row) and a combination of all uncertainties (fourth row). TCCON observations, when available, are shown for the 5 days (black circles).**





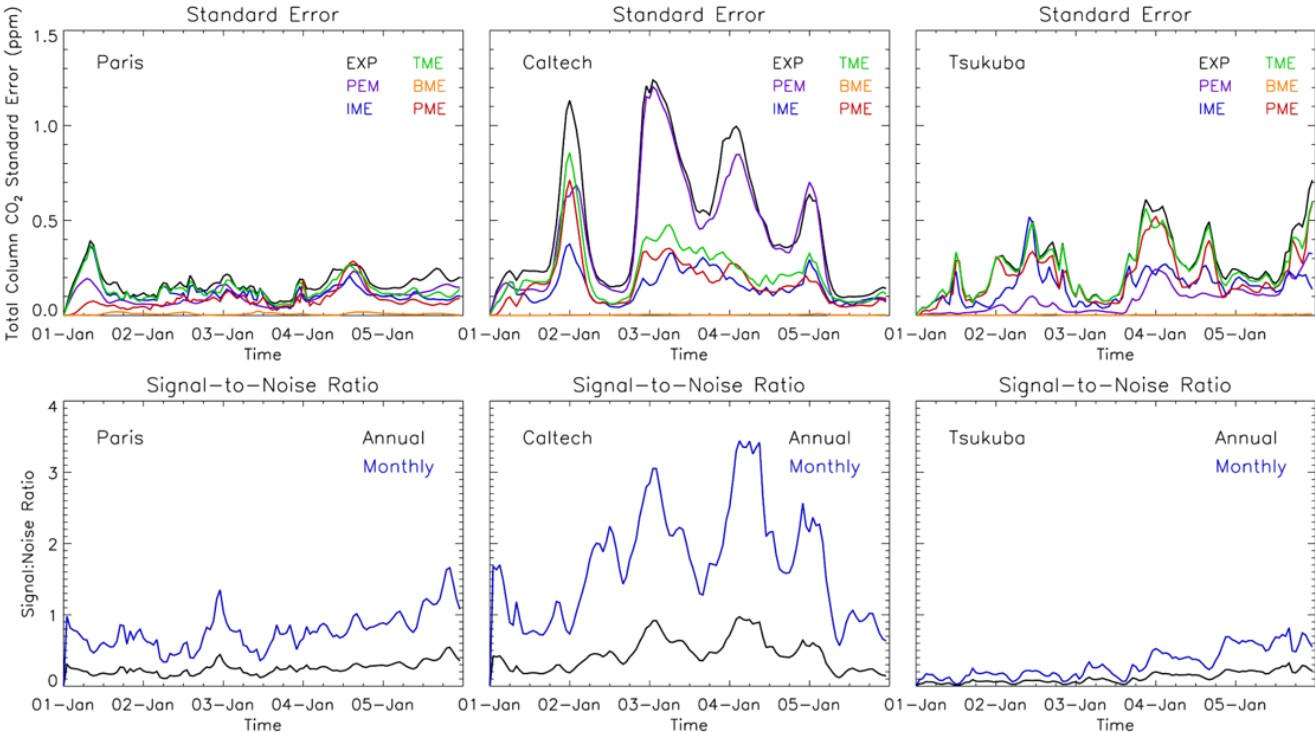

**Figure 2. IFS model XCO₂ (ppm) standard error across 50-member ensemble over three TCCON sites for 7 different model configurations (top row). The XCO₂ signal generated by uncertainties in anthropogenic emissions divided by the noise from remaining model error over the same TCCON sites (bottom row).**




**Figure 3. Standard error of IFS model CO₂ profiles (ppm) across 50-member ensemble for 1st January 2015 over three TCCON sites. Ensemble configurations consist of perturbed initial meteorological conditions (top row), perturbed model physics (second row), both perturbed initial conditions and physics (third row), perturbed biogenic emission caused by transport uncertainty (fourth row), perturbed emissions using monthly anthropogenic uncertainties per sector and country (fifth row), perturbations of the combined transport, biogenic feedback and anthropogenic emission uncertainties (bottom row).**






**Figure 4. Global standard error of IFS model XCO$_2$ (ppm) across 50-member ensemble after 6 hours, 24 hours and 10 days. Errors shown are from uncertainties in biogenic emissions caused by meteorological uncertainty (top left), monthly anthropogenic emission uncertainties per sector and country (top right), model transport uncertainty (bottom left) and a combination of all uncertainties (bottom right).**



**Figure 5.** Global signal-to-noise ratio of IFS model XCO2 across 50-member ensemble after 6 hours (top), 24 hours (middle) and 10 days (bottom), where the signal is the atmospheric response to annual (left) and monthly (right) anthropogenic emission uncertainty and the noise is the transport and biogenic feedback error.





**Figure 6. Global average XCO₂ (ppm) standard error from IFS model over 15 days for a 3 (red), 5 (orange), 10 (green), 15 (turquoise), 25 (blue), 40 (purple) and 50 (black) member ensemble (top). A binned density plot of the change in normalised error, relative to the 50-member ensemble, with respect to ensemble size (second row). The normalised error is computed for each ensemble size for 120 different times (January 2015) before being binned. Histogram showing IFS model XCO₂ from a 10, 25 and 50 member ensemble after 5 days. Note that all ensembles shown consist of initial meteorological uncertainty and perturbed model physics (TME).**




**Figure 7. A snapshot of regional XCO₂ error correlation structure with respect to Paris XCO₂ from 10 (top left) and 50 (top right)**
**member IFS model ensemble after 4 days, where the ensemble consists of perturbed initial meteorology and model physics (transport error). The middle-row shows the same as the top-row but includes the preceding and subsequent model time steps (±1 hour), artificially increasing the correlation sample to 30 and 150 members. The bottom row shows the same correlation calculations as the middle-right panel (150-member consisting of ±1 hour) but for two different times; highlighting the flow-dependence in error correlation structure. The star denotes the column over Paris and the black arrows denote the down- and across-wind directions**
**used to calculate the further and shortest correlation lengths for a given time (see Figure 8).**





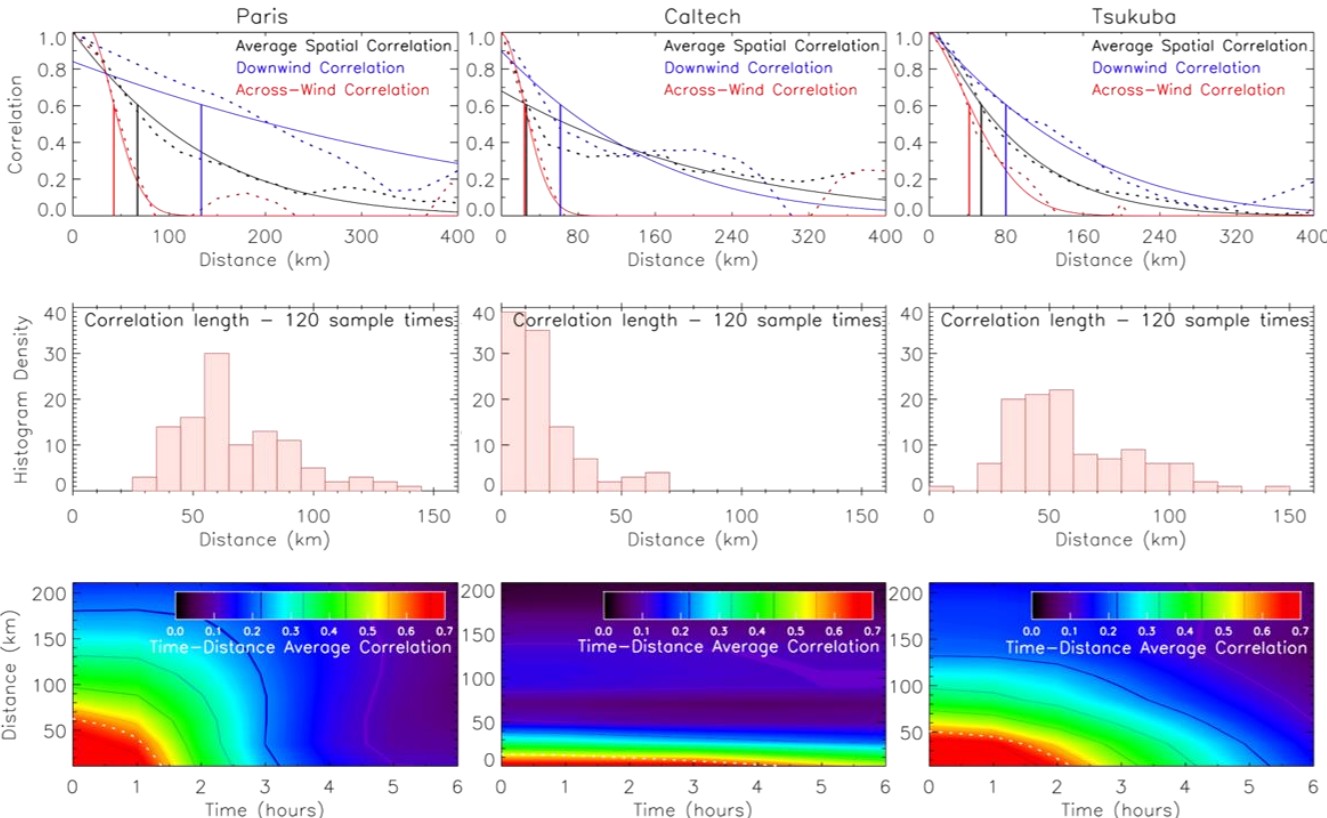

**Figure 8. A snapshot of XCO2 error correlation with respect to Paris (left), Caltech (middle) and Tsukuba (right) as a function of distance for a 50-member IFS model ensemble after 4 days (top row). These panels show the directionally averaged (black dashed line), downwind (blue dashed line) and across-wind (red dashed line) correlation values are shown with a gaussian fit (solid lines) in addition to the derived correlation length where $R = e^{-0.5}$ (vertical solid lines). The directionally averaged derived correlation lengths for 120 sample times for January 2015 are placed in 10km bins for all three sites (middle row). The directionally and time averaged error correlation values for the same 120 sample sizes as a function of both time and distance (bottom row).**