# Peer review of "Representing Model Uncertainty for Global Atmospheric CO2 Flux Inversions Using ECMWF-IFS-46R1"

_Geoscientific Model Development, 2019_

## Referee Comment (RC1) · Anonymous Referee #1 · 13 Feb 2020

This paper describes the detailed uncertainties in atmospheric CO2 estimates based on inversions from a particular ECMWF weather prediction model. This type of un-certainty analysis of model products is important and useful. The model experiment setup is well conceived and the results may help in the uncertainty analysis of CO2 inversions from other models and meteorology. The main issue I have with the paper is in some of the results discussion that is too heavy on citing numbers and some figures that need to be made more clear and have more discussion of their specific features. Specific comments are listed below. In general, the topic of this paper is appropriate for Geoscientific Model Development and I recommend publication provided the results discussion and figures can be made more focused and easily readable.

[Figure]

Specific comments

Lines 37 and 272: 'typically' instead of 'typical'

Lines 60-64: This paragraph ("Here, we investigate...") seems like it should be in the abstract. I understood better and more quickly what you are doing in the paper after reading this paragraph compared to any part of the abstract. You might consider removing some of the specific numbers and results in the abstract and replacing it with this paragraph and perhaps also some of the sentences in lines 103-107.

Section 4.1: This section is a bit hard to read. I got bogged down with all of the citing of error numbers, trying to find the quoted errors on the plots and looking for descriptions of some of the details of Figures 1 and 2. Maybe the monthly mean errors could be put in a table so they don't have to be listed in the text? The acronyms aren't always consistent between the labeling on the Figures and in the text. The eight different model experiments and acronyms are a lot to keep track of so consistency and repetition helps the reader.

Figure 1: I understand each of the colored lines represent individual ensemble members but I was initially trying to figure out if there was any information in the red lines vs. the blue or green lines since that's generally the case in line plots with multiple colors. An alternative coloring scheme that would seem to emphasize the main point of the figure and connect it better to Fig. 2 is to color all of the ensemble members the same shade of gray or some other color, then add an average of the 50 ensembles with a thicker line and darker color along with the standard error at each time as whiskers on the mean. The standard error whiskers would then correspond to the lines on Fig. 2 for each model case.

It would be helpful to label the plots using the same acronyms as in Table 1 and Figures 2 and 3. What are the numbers at the bottom of each plot? I don't see any mention of them in the caption or text.

Figure 2: It's hard to differentiate between the darker colors in the top row of plots. Does IME+PME=TME?

Figure 3 and lines 267-71: The maximum values mentioned in the text are much larger than the color scale on Figure 3. Maybe a logarithmic scale would be better?

―――――――――――

---

## Referee Comment (RC2) · Anonymous Referee #2 · 17 Feb 2020

Overview: The manuscript "Representing Model Uncertainty for Global Atmospheric CO2 Flux Inversions Using ECMWF-IFS-46R1" by McNorton et al. describes the quantification of various types of transport and emission errors when simulating atmospheric CO2 using an online earth system model. A number of experiments were carried out using an ensemble of 50 model simulations, which were perturbed in such a way as to be able to quantify various components of transport error, errors in the feedback to biogenic fluxes of CO2, and anthropogenic flux errors. This allowed for the estimation of a signal-to-noise ratio within the model. An attempt to account for spurious spatial error correlations was also included, through filtering of the model output in time. The paper suggests that model transport errors are a significant source of error for flux inversions,

and should be more carefully accounted for in general.

Overall the manuscript is very well written, with few technical corrections necessary. Some details are skipped over without thorough explanation, however. The figures are generally quite clear and well chosen, although some further detail needs to be provided for some of them. The methods and models used within the manuscript are appropriate for such a study, and the study overall provides some idea of the size of model transport errors relative to emission errors for the two periods studied. The methods present a smart, if computationally-intensive, blueprint for other modellers to estimate model errors in a CCM setting.

I have few reservations regarding the publication of this manuscript in GMD, subject to some relatively minor changes, detailed below. Once these are fixed, I suggest that this paper is suitable for publication in this journal.

Comments:

Page 6, paragraph beginning at line 160: I gather that descriptions of the emission uncertainty experiments EXP, PEM and PEA are given in a forthcoming manuscript, and briefly here in Table 1. However, an extra sentence or two to briefly describe them, and the general differences between annual and monthly error, would be useful here. I later understood these experiments through context.

Page 8, line 199: To be clear, the two day spin-up is the 1st and 2nd of each month, included in some of the figures, and not the 30th and 31st of the previous month? Please make this clear in the text.

Figure 1: What do the numbers in these panels represent? If they represent model spread, what times do they represent?

Figure 2: Some comment on the fact that PEM is sometimes greater than EXP, and what this means, should be included in the main text.

Page 9, line 245: When discussing July, only one S/N ratio is provided. I assume this

is the monthly version? This should be stated. This is also true in Figure S2.

Page 10, line 263: The 24-hour period is January 1st (which should be stated). My understanding is that this date is discarded as spin-up when deriving monthly uncertainties. Would it be better to show a later date here and in Figure 3?

Page 10, line 283: In this section, the global (and monthly?) average error is often provided. I'm not sure whether this is a useful diagnostic since the error is very heterogeneous for the flux error cases. More helpful would be to provide values for a few of the affected regions, as is done for the transport error case from line 296. This would allow the reader to compare the biogenic and transport errors over the Amazon, for example.

Page 11, line 305: This exact sentence is already included earlier and I assume that it has been moved?

Page 13, line 324: What do you mean by 'as a first guess'?

Page 14, lines 346-351: I find these lines a little confusing, and they could be more clearly rewritten. I think you're saying that non-spurious error correlations can be found at varying - and surprising large - distances from a particular grid cell, and that using predefined localisation scales could remove error correlation information? It should be mentioned that this is difficult to account for in real inversions without thorough prior assessment of model output, isn't it? Also, how do you define whether correlations are 'part of the spatial extent of the plume'?

Page 15, line 355: How do you assess what is 'downwind'? Through analysis of the model winds? Or inspection of the plume? Is it instantaneous or averaged?

Abstract and line 431: You should clarify how your results can be used in future offline studies by the inversion community. Are you suggesting that modellers use the diagnosed transport errors derived here to drive their own offline inversions? Is the intention to produce these fields for periods other than January and July 2015?

Technical corrections:

Page 1, line 8: "prior flux, a forward model" rather than "prior flux, forward model".

Page 5, line 146: Remove comma after '(TME)'.

Page 6, line 173: comma after 'principle'.

---

## Author Comment (AC1) · 18 Mar 2020

**Authors Response to Reviews**

We would like to thank both reviewers for their insightful and constructive feedback on the manuscript. We have addressed all comments below and in an updated version of the text.

**Response to referee #1**

This paper describes the detailed uncertainties in atmospheric CO2 estimates based on inversions from a particular ECMWF weather prediction model. This type of uncertainty analysis of model products is important and useful. The model experiment setup is well conceived and the results may help in the uncertainty analysis of CO2 inversions from other models and meteorology. The main issue I have with the paper is in some of the results discussion that is too heavy on citing numbers and some figures that need to be made more clear and have more discussion of their specific features. Specific comments are listed below. In general, the topic of this paper is appropriate for Geoscientific Model Development and I recommend publication provided the results discussion and figures can be made more focused and easily readable.

We would like to thank the reviewer for their comments, we acknowledge that the discussion numbers are at times too heavy and the explanation of the figures is sometimes lacking. We have addressed these issues within the text through our responses to each comment listed below.

Lines 37 and 272: 'typically' instead of 'typical'

This has now been updated in the text.

Lines 60-64: This paragraph ("Here, we investigate. . .") seems like it should be in the abstract. I understood better and more quickly what you are doing in the paper after reading this paragraph compared to any part of the abstract. You might consider removing some of the specific numbers and results in the abstract and replacing it with this paragraph and perhaps also some of the sentences in lines 103-107.

We agree this section would be useful in the abstract. We have updated the abstract using some text from the suggested parts of the main body.

Section 4.1: This section is a bit hard to read. I got bogged down with all of the citing of error numbers, trying to find the quoted errors on the plots and looking for descriptions of some of the details of Figures 1 and 2. Maybe the monthly mean errors could be put in a table so they don't have to be listed in the text? The acronyms aren't always consistent between the labeling on the Figures and in the text. The eight different model experiments and acronyms are a lot to keep track of so consistency and repetition helps the reader.

We agree that this section was messy, we have made the suggested edits by removing values from the text and placing them into a new table (now table 2). We have also ensured consistent labelling between the figures and text.

Figure 1: I understand each of the colored lines represent individual ensemble members but I was initially trying to figure out if there was any information in the red lines vs. the blue or green lines since that's generally the case in line plots with multiple colors. An alternative coloring scheme that would seem to emphasize the main point of the figure and connect it better to Fig. 2 is to color all of the ensemble members the same shade of gray or some other color, then add an average of the 50 ensembles with a thicker line and darker color along with the standard error at each time as whiskers

on the mean. The standard error whiskers would then correspond to the lines on Fig. 2 for each model case. It would be helpful to label the plots using the same acronyms as in Table 1 and Figures 2 and 3. What are the numbers at the bottom of each plot? I don't see any mention of them in the caption or text.

We agree Figure 1 was inconsistent with the other figures and have updated the labelling, so that it fits better with both the text and other figures. We have also adopted an alternative colour scheme as suggest where the solid black line represents the mean and each member is a lighter grey line. We have now also explained the values given in the plots.

**Figure 2: It's hard to differentiate between the darker colors in the top row of plots. Does IME+PME=TME?**

We have now adjusted the colours slightly so the dark purple and blue are both lighter to provide more contrast. IME + PME does equal TME for the purpose of the experimental setup, however the standard error when the two are combined is not necessarily equal to the sum of the two added.

*Figure 3 and lines 267-71: The maximum values mentioned in the text are much larger than the color scale on Figure 3. Maybe a logarithmic scale would be better?*

We agree that the scale was not appropriate and have updated the figure with a logarithmic scale as suggested.

**Response to referee #2**

Overall the manuscript is very well written, with few technical corrections necessary. Some details are skipped over without thorough explanation, however. The figures are generally quite clear and well chosen, although some further detail needs to be provided for some of them. The methods and models used within the manuscript are appropriate for such a study, and the study overall provides some idea of the size of model transport errors relative to emission errors for the two periods studied. The methods present a smart, if computationally-intensive, blueprint for other modellers to estimate model errors in a CCM setting.

I have few reservations regarding the publication of this manuscript in GMD, subject to some relatively minor changes, detailed below. Once these are fixed, I suggest that this paper is suitable for publication in this journal.

We would like to thank the reviewer for their comments, we acknowledge that some details were lacking. We have addressed these issues within the text through our responses to each comment listed below.

Page 6, paragraph beginning at line 160: I gather that descriptions of the emission uncertainty experiments EXP, PEM and PEA are given in a forthcoming manuscript, and briefly here in Table 1. However, an extra sentence or two to briefly describe them, and the general differences between annual and monthly error, would be useful here. I later understood these experiments through context.

We agree the details are lacking within the manuscript and they will indeed be given in an upcoming manuscript submitted to ESSD. We agree that more information regarding the derivation of uncertainties is required and have updated the text to include more details.

Page 8, line 199: To be clear, the two day spin-up is the 1st and 2nd of each month, included in some of the figures, and not the 30th and 31st of the previous month? Please make this clear in the text.

We agree this is a point that should be clarified in the text and have updated the text to include the spin-up is for the  $1^{st}$  and  $2^{nd}$  of the respective month.

**Figure 1: What do the numbers in these panels represent? If they represent model spread, what times do they represent?**

We have updated the caption to include a description of the values as commented on by referee #1. Values are model spread for different times.

**Figure 2: Some comment on the fact that PEM is sometimes greater than EXP, and what this means, should be included in the main text.**

We thank the referee for their feedback on this and are unsure what is causing this. We have added in a coupe of hypothesis within the text that the larger error seen for brief periods in PEM over Caltech may either be due to spurious noise from the small ensemble size or by compensating errors from other sources e.g transport/biogenic causing a reduction in the total error.

*Page 9, line 245: When discussing July, only one S/N ratio is provided. I assume this is the monthly version? This should be stated. This is also true in Figure S2.*

This is indeed the monthly version of the S/N ratio, we have updated the text to state this.

Page 10, line 263: The 24-hour period is January 1st (which should be stated). My understanding is that this date is discarded as spin-up when deriving monthly uncertainties. Would it be better to show a later date here and in Figure 3?

We agree that using the first 24 hours as depicted in Figure 3 is not suitable, we have updated the figure for a 24 hour period on day 5.

Page 10, line 283: In this section, the global (and monthly?) average error is often provided. I'm not sure whether this is a useful diagnostic since the error is very heterogeneous for the flux error cases. More helpful would be to provide values for a few of the affected regions, as is done for the transport error case from line 296. This would allow the reader to compare the biogenic and transport errors over the Amazon, for example.

We agree that the global average is not a very useful diagnostic that is why we now focus the results on specific sites, this is shown for both the three TCCON sites but also the emission hotspots from table 3 and 4. We have included the comment that the global values are heterogenous and emphasised hotspots are where the signal is most detectable.

Page 11, line 305: This exact sentence is already included earlier and I assume that it has been moved?

This has now been removed from the text.

Page 13, line 324: What do you mean by 'as a first guess'?

We agree this is an ambiguous statement with the intention that this is an early stage approach to investigating ensemble sizes and further studies may want to consider more sophisticated approaches. For clarity we have removed this phrase from the text.

Page 14, lines 346-351: I find these lines a little confusing, and they could be more clearly rewritten. I think you're saying that non-spurious error correlations can be found at varying - and surprising large - distances from a particular grid cell, and that using predefined localisation scales could remove error correlation information? It should be mentioned that this is difficult to account for in real inversions

**without thorough prior assessment of model output, isn't it? Also, how do you define whether correlations are 'part of the spatial extent of the plume'?**

We agree this section is somewhat confusing, we have re-written parts of the text to clarify the key points. We have also emphasised the difficulty of using full flow-dependent error structures in offline inversions and highlighted that simplification may be required. We have detailed the spatial extent as being a length along a direction from the location which has error correlations continuously above  $e^{-0.5}$ .

**Page 15, line 355: How do you assess what is 'downwind'? Through analysis of the model winds? Or inspection of the plume? Is it instantaneous or averaged?**

We define downwind as being along the plume direction, we checked this with wind direction and it was found to be typically the same direction, therefore we have kept the term downwind to encompass both the plume and wind direction. The direction is instantaneous at the model output time, we have updated the text to clarify this.

Abstract and line 431: You should clarify how your results can be used in future offline studies by the inversion community. Are you suggesting that modellers use the diagnosed transport errors derived here to drive their own offline inversions? Is the intention to produce these fields for periods other than January and July 2015?

We agree and an aim for the study is that the transport errors can be used by the wider inverse modelling community. We have updated the text to state global standard errors are available upon request for the total column mixing ratios at 3 hourly intervals for 2015 and both total column and surface errors at hourly intervals for January and July 2015. Although not expressly stated other errors reported in the study can be made available upon request.

In addition to this we have created a Zenodo online dataset for the data with a doi and added this information into the data availability section.

Page 1, line 8: "prior flux, a forward model" rather than "prior flux, forward model".

Page 5, line 146: Remove comma after '(TME)'.

Page 6, line 173: comma after 'principle'.

All the above are now corrected.